# The Phenolic Compounds’ Role in Beer from Various Adjuncts

**DOI:** 10.3390/molecules28052295

**Published:** 2023-03-01

**Authors:** Irina N. Gribkova, Mikhail N. Eliseev, Irina V. Lazareva, Varvara A. Zakharova, Dmitrii A. Sviridov, Olesya S. Egorova, Valery I. Kozlov

**Affiliations:** 1All-Russian Scientific Research Institute of Brewing, Beverage and Wine Industry—Branch of V.M. Gorbatov Federal Research Center for Food Systems, 119021 Moscow, Russia; 2Academic Department of Commodity Science and Commodity Expertise, Plekhanov Russian University of Economics, 117997 Moscow, Russia

**Keywords:** beer, organic compounds, adjunct, phenolic compounds, beer’s proteome, test descriptors

## Abstract

Background: The present article considers the influence of malt with various adjuncts on beer organic compounds and taste profile composition, with more attention paid to the phenol complex change. The topic under consideration is relevant since it studies the interactions of phenolic compounds with other biomolecules, and expands the understanding of the adjuncts organic compounds contribution and their joint effect on beer quality. Methods: Samples of beer were analyzed at a pilot brewery using barley and wheat malts, barley, rice, corn and wheat, and then fermented. The beer samples were assessed by industry-accepted methods and using instrumental analysis methods (high-performance liquid chromatography methods—HPLC). The obtained statistical data were processed by the Statistics program (Microsoft Corporation, Redmond, WA, USA, 2006). Results: The study showed that at the stage of hopped wort organic compounds structure formation, there is a clear correlation between the content of organic compounds and dry substances, including phenolic compounds (quercetin, catechins), as well as isomerized hop bitter resines. It is shown that the riboflavin content increases in all adjunct wort samples, and mostly with the use of rice—up to 4.33 mg/L, which is 9.4 times higher than the vitamin levels in malt wort. The melanoidin content in the samples was in the range of 125–225 mg/L and its levels in the wort with additives exceeded the malt wort. Changes in β-glucan and nitrogen with thiol groups during fermentation occurred with different dynamics and depending on the adjunct’s proteome. The greatest decrease in non-starch polysaccharide content was observed in wheat beer and nitrogen with thiol groups content—in all other beer samples. The change in iso-α-humulone in all samples at the beginning of fermentation correlated with a decrease in original extract, and in the finished beer there was no correlation. The behavior of catechins, quercetin, and iso-α-humulone has been shown to correlate with nitrogen with thiol groups during fermentation. A strong correlation was shown between the change in iso-α-humulone and catechins, as well as riboflavin and quercetin. It was established that various phenolic compounds were involved in the formation of taste, structure, and antioxidant properties of beer in accordance with the structure of various grains, depending on the structure of its proteome. Conclusions: The obtained experimental and mathematical dependences make it possible to expand the understanding of intermolecular interactions of beer organic compounds and take a step toward predicting the quality of beer at the stage of using adjuncts.

## 1. Introduction

The structure of food products, including beverages, is formed on the basis of the interaction of organic compounds, which are contained in raw materials and can be extracted as a result of the conditions of the technological cycle. In this regard, the beer colloidal structure consists of primary, biomodified, and newly formed organic structures of plant raw materials: Cereal, hop-based, as well as obtained through the microorganism’s cultures [1,2].

The focus of our research interest is on the phenolic compounds of plant raw materials, which are still the most studied in terms of the sources of turbidity formation as well as nutraceutical properties [3,4,5,6].

Among the phenolic compounds, simple phenolic acids and aldehydes, catechins, proanthocyanins, prenylated flavonoids α- and iso-α-acids, etc., have been described in beer [6]. The listed classes of compounds correlated with the styles of beer: Prenylated flavonoids were most abundant in dark stout beer (3.10 mg/L) and dry-hopped beer (3.68 mg/L), while alkylresorcinols were more abundant in stout beer (4.52 μg/L) [6]. Alkylresorcinols belong to a group of phenolic lipids of cereals (barley, wheat, rye, oats, rice, and other cereal grains), and contain resorcinol [7,8]. Other phenolic compounds, such as tyrosol and hydroxytyrosol, are yeast products and derived from tyrosine [9]. In addition, their concentration in beer was 0.2–44.4 and 0.0–0.1 mg/L.

Therefore, there are phenolic compounds of cereal sources (malt or unmodified grain raw material), hop origin, as well as those resulting from the microorganism’s activity.

Non-fermented grains and malt in beer are sources of flavan-3-ols, proanthocyanidins, hydroxycinnamic acid derivatives, and small amounts of flavanols [10] in bound and free forms. It is widely believed that catechin and ferulic acid [11] are the most common flavonoid compounds. Moreover, it has been shown that hydrolytic processes of dissolution, fermentation, and drying of malt endosperm lead to a decrease in the content of catechin, prodelfinidin B3, procyanidin B3, and ferulic acid in the grain [12,13] and an increase in the content of water-soluble esterified fractions of phenolic compounds [13,14]. Elevated temperatures in the final stage of commercial malt production lead to a decrease in the amount of ferulic acid included in the structure of melanoidins, which indicates the esterification of acids and nitrogenous compounds of the cereal matrix [15,16]. Furthermore, malt is a source of mono-O-glucosides (myricetin and myricetin-O-glucoside) [17].

Phenolic compounds of native cereals are rarely detected in a free form since they are esterified or glycosylated with organic compounds of cereals (lignin, β-glucans, arabinoxylans, nitrogenous compounds, and other organic compounds of the plant matrix) [18]. Table 1 shows the composition of cereal phenolic compounds and their possible presence in beer.

The main hop polyphenols are flavonols (quercetin and kaempferol), flavan-3-ols (catechins, epicatechin, proanthocyanidins), phenolic acids, and prenylated chalcones. Prenylated flavonoids combine the flavonoid skeleton with a lipophilic prenyl side chain, increasing the lipophilicity of flavonoids [41,42]. Isoxanthohumol is the main prenylated chalcone in lupulin glands (0.1–1% in dry weight); however, during the brewing process, it isomerizes the 2’-hydroxy group and is converted to isoxanthohumol (ranging from 0.04 to 3.44 mg/L), which also contributes to the bitterness of beer [43,44].

As a result of temperature, pressure, oxygen, and other conditions, some phenolic compounds are oxidized, biotransformed, and precipitated, which affects the structure and sensory perception of the beer. The effect of phenolic compounds can be direct or indirect. Therefore, hop products introduced after fermentation provided an extraction of humulon and lupulon, while oxidation of those in the container provided a “smoky” tone, which is an off-flavor [45]. The indirect effect is due to biotransformation or decarboxylation of phenolic compounds by yeast enzymes to 4-vinylguaiacol, which is responsible for the clove aroma [46]. In addition, the taste and aroma tones depend on the concentration of the compound [47]. It has been shown that isoferulic acid causes fruit tones [48]. The direct effect of phenolic compounds is associated with their antioxidant properties that counteract the oxidation of other organic compounds [49].

Table 2 shows the beer phenolic compounds from various cereal raw materials (adjuncts) [50,51,52].

As shown in Table 2, many classes of phenolic compounds are lost during technological processes and are not present in the beer, which is also confirmed by other authors [53].

Table 3 shows the flavor attributes according to the adjuncts used.

Therefore, the role of phenolic compounds and their influence on beer quality is extensive and depends on the raw materials used, technological parameters of production, storage conditions, etc. As a result, the purpose of the study was to establish implicit relationships between beer organic compounds produced from different adjunct types and assess their impact on beer quality.

## 2. Results

### 2.1. The Determination and Mathematical Analysis of Different Adjuncts on Wort Sample Composition

The wort samples with different adjuncts studied in this work were selected in order to reveal the difference in the influence of various phenolic and other compounds of plant raw materials on the hopped wort flavor descriptors.

Table 4 and Table 5 show the composition of the wort samples.

In Table 4, the data indicate that the addition of different adjuncts affects the quantitative expression of the compounds responsible for the formation of the beer’s structure. The β-glucan content increased in relation to the malt wort level by a factor of 1.3, 1.6, 1.5, 1.8, and 2.9 in barley malt + barley, barley malt + rice, barley malt + corn, wheat malt + wheat wort samples, accordingly. Moreover, the content of soluble nitrogen was 1.1, 1.3, 1.2, and 1.25 times higher in the wort with the adjuncts’ addition when barley, rice, corn, and wheat were added, respectively. The wort obtained from wheat malt and wheat contains the least soluble nitrogen, which correlates with the wort original extract (10.2 °P). The content of nitrogen (peptones) with thiol groups does not correlate with the wort content of soluble nitrogen and original extract. Note that the content of iso-α-humulone in the wort correlates with the content of the wort original extract, since a similar amount of the same hop species was applied in all cases under consideration.

In Table 5, the data indicate the influence of the composition of grain raw materials on the hopped wort matrix structure. In fact, the quantitative composition of the matrix structure of organic compounds is determined by the adjuncts’ composition.

The higher content of catechins was observed in wheat wort (81.7 mg/L) and the lowest in barley malt wort (18.6 mg/L). The melanoidin’s content was naturally higher in the wort with adjuncts, except for wort with barley malt + barley. Here, there is an increase in free amino nitrogen content, which occurs when the protein matrix undergoes proteolysis during mashing and the thermal reaction of sugars and amino acids during temperature pauses. Moreover, the riboflavin content is increased in the wort with adjuncts and mostly in barley malt + rice wort samples (4.33 mg/L), which is 9.4 times higher than the barley malt wort.

In Table 5, the data were subjected to mathematical processing and obtained from pairwise correlation coefficients, and thus describe the strength of the relationship between the compounds, as shown in Table 6.

Pairwise correlation coefficients (Table 6) showed that catechins have a direct effect on color to the greatest extent. Melanoidins along with riboflavin have a strong influence on color (pairwise correlation coefficient of 0.87), as well as catechins along with riboflavin are associated with and determine the color characteristics formation (pairwise correlation coefficient of 0.91). The overall system correlation coefficient under consideration is R = 0.96, and the coefficient of determination is R^2^ = 0.94, which indicates that an additional effect on color formation of 6% was unaccounted for in the compounds.

### 2.2. The Variation of Different Adjuncts on Wort Sample Composition during Brewing

To identify implicit connections of the colloidal structure of organic compounds, which are formed during wort fermentation with different adjuncts, we monitored the dynamics of changes in the significant compounds content, which is reflected in Figure 1a–d.

The change in the content of β-glucan (Figure 1a) occurs most significantly for a total of three times in fermented wort from wheat malt + wheat, since the formation of colloidal particles consisting of nitrogenous, phenolic, and non-starch carbohydrate compounds occurs, which are sufficiently high in molecular weight to remain in equilibrium in the dispersed structure [66].

The decrease in the concentration of non-starch polysaccharide is more linear in the other cases with a decrease in β-glucan content observed by 1.1–2 times. Moreover, we can arrange the options of cereal composition in the following order by a decrease in β-glucan content during fermentation: Wheat malt + wheat→ barley malt + rice→ barley malt + barley → barley malt + corn→ barley malt fermented wort samples. On the one hand, a lower β-glucan concentration indicates a more dissolved structure of the cereal components and a lower molecular weight of the non-starch polysaccharide inherent in the fermented wort; on the other hand, this is indicated by the type of structure [67,68].

Changes in soluble nitrogen (peptones) with thiol groups (Figure 1b) show interesting dynamics and divisions of samples into groups. Based on the changes in nitrogen compounds, natural barley malt + corn, barley malt + barley, barley malt + rice beer samples can be combined into the first group, and wheat malt + wheat, barley malt + wheat beer samples can be combined into the second group. The first group of samples is characterized by a high intensity of nitrogen reduction in the first 7 days of fermentation and a smooth decrease in the following days, whereas the second group is characterized by a systematic reduction in nitrogen with thiol groups during the whole period of fermentation. If beer samples are ranked quantitatively with respect to nitrogen compounds with thiol groups at the beginning of fermentation, then the series would be as follows: Barley malt + corn→ barley malt + barley→ barley malt→ wheat malt + wheat→ barley malt + rice→ barley malt + wheat beer samples. By the end of fermentation, a different trend was observed: Wheat malt + wheat→ barley malt + corn→ barley malt + barley→ barley malt + wheat→ barley malt→ barley malt + rice beer samples. Therefore, the use of malted and non-malted wheat contributes to a smaller decrease in thiol-containing nitrogen compounds in the beer: In the case of wheat malt + wheat beer samples, the thiol-containing nitrogen compounds content decreased by 4 times (up to 70.2 μM/L), and in the case of barley malt + wheat beer samples, it decreased by 7.3 times (up to 29.8 μM/L), which is explained by the presence of nitrogenous fraction in the beer structure from modified and non-modified wheat raw materials with thiol groups of molecular mass 2.1–40 kDa, which are responsible for the foam quality [69].

The content of catechins (Figure 1c) changed typically in all samples, except for wheat malt + wheat beer samples, which contained the greatest catechins amount (81.68 mg/L) and in 7 days, it decreased to 10.4 mg/L, i.e., 8 times, which distinguished this sample from all others, in which the catechins content decreased equally. It should be noted that the catechins content in the wort was in the range of 18.56–81.68 mg/L, while during the fermentation this indicator decreased by 1.7–12.8 times. The highest decrease in these phenolic compounds was shown by the sample obtained from barley malt and barley, and the lowest by the sample from barley malt and corn. This corresponds to an inversely proportional relationship with the quantitative characteristic of the catechins content in non-malted adjuncts, with the highest amount contained in barley (46.9 μg/g), followed by rice (13.7 μg/g), then corn (1.7 μg/g), and wheat (0.008 μg/g) [19,20,21,22,23,24,25,26,27,28,29,30,31,32,33,34,35,36,37,38,39,40,70]. Moreover, it is necessary to take into account the structure, the structural relationships in the plant matrix of individual cereals, and the conditions of malt and wort production [71,72].

Regarding the iso-α-humulone content (Figure 1d), we can conclude that the characteristics of changes in the isomerized resin amount are identical in all samples, namely, it does not depend on the adjunct’s type, but more on the original extract content. The highest amount of iso-α-humulone is in wheat malt + wheat fermented wort samples with an original extract content of 10.2 °P, while in other samples of fermented wort, the iso-α-humulone amount is 11.0 °P and above. It was noted that according to the data obtained, the content of iso-α-humulone correlates with the content of dissolved substances in the wort, as in the case of wheat malt + wheat wort samples. However, there is no correlation in the fermented wort. Apparently, this is due to implicit connections with other organic compounds in the beer’s structure.

The quercetin content in the wort samples correlates with its content in the cereal [19,20,21,22,23,24,25,26,27,28,29,30,31,32,33,34,35,36,37,38,39,40], but this correlation is not observed in the fermented wort. The highest quercetin content is in fermented wheat malt + wheat beer samples (0.66 mg/L), followed in descending order by the beer samples with barley malt and barley→ wheat→ rice→ 100% malt beer→ corn.

The mathematical correlation multivariate analysis method was applied to identify implicit relationships between the change in nitrogen content with thiol groups (Y), catechins (X_1_), quercetin (X_2_), and iso-α-humulone (X_3_) based on data from Table 4 and Table 5 and Figure 1, as shown in Table 7 and Table 8.

Barley malt beer sample (Table 7) was characterized by high values of pairwise correlation coefficients, indicating all organic compounds under consideration (R = 0.86–0.99). The highest correlation coefficient (strong connection) was characterized by the quercetin and nitrogen compounds with thiol groups connection. The same pattern was observed in barley malt + barley and barley malt + rice beer samples, as well as in barley malt beer, and the pairwise correlation coefficient between quercetin and thiol nitrogen compounds was R = 0.98. Barley malt + corn beer samples had lower values of pairwise correlation coefficients (R = 0.59–0.75) compared to the above samples, and barley malt + wheat and wheat malt + wheat beer samples had values of R = 0.92–0.99 and 0.65–0.95, respectively, indicating the participation of unaccounted organic compounds in the relationship between quercetin, catechins, iso-α-humulone, and nitrogen compounds with thiol groups.

Pairwise correlation coefficients (Table 8) showed a strong mutual influence of phenolic compounds on thiol nitrogen compounds in all beer samples regardless of adjuncts type. The elasticity coefficient characterizing the degree of influence of one parameter on the other showed the significance of iso-α-humulone and catechin with respect to the connection with thiol nitrogen compounds. Therefore, it was interesting to assess the presence or absence of pairwise correlation coefficients between these indicators in different beer samples.

In Figure 1, the data were mathematically analyzed for a linear relationship between iso-α-humulone and catechins, and the data are presented in Table 9.

It is possible to describe the equations in Table 9 mathematically by the coefficient before the variable X, denoting the tangent of the slope angle: The greater the tangent of the angle, the more vigorous the binding reaction of catechins and iso-α-humulone [73]. The highest coefficient is characterized by barley malt + corn beer samples, and then follows in the degree of rate reduction: Barley malt→ barley malt + rice→ barley malt + barley→ barley malt + wheat→ wheat malt + wheat beer samples. Note that the free coefficient characterizes the minimum value of iso-α-humulone, at which its equilibrium reacts with catechin coupling. Moreover, the lower the free coefficient, the higher the association reaction rate.

Table 6 shows the beer’s color formation, in which catechin and riboflavin molecules are significantly associated with beer color. However, it was interesting to observe whether there was a connection between riboflavin and quercetin. For this purpose, the data obtained during fermentation were mathematically processed, and the results are presented in Table 10.

In Table 10, the data indicate the highest coefficient before the variable is characterized by barley malt + corn beer samples, followed by: Barley malt + barley→ barley malt→ barley malt + wheat→ wheat malt + wheat→ barley malt + rice beer samples.

Based on the data obtained (Table 9 and Table 10), we plotted the dependences of changes in iso-α-humulone content on catechins (Figure 2a) and riboflavin content on quercetin (Figure 2b).

The graphs and dependences of Figure 2 and Table 10 are linear in nature, and the strength of the relationship between the variables, the change in which occurs according to these patterns, is close to functional in nature.

### 2.3. The Matrix Organic Compounds Influence on Beer’s Descriptor Flavor Formation

In Table 11, the data indicate the beer’s characteristics using different types of adjuncts, and Figure 3 shows the taste profiles of beer samples.

It is interesting to note that, regarding the data in Table 11, the highest β-glucan content was in the beer sample obtained from 95% barley malt and 5% wheat—the content of non-starch polysaccharide exceeded the content of barley malt beer sample by 2.3 times. Moreover, there was an increase of 25%, 78%, 7%, and 43% in β-glucan content in barley malt and barley, rice, corn, and wheat beer samples, respectively. The soluble nitrogen compound content was in range of 495–695.6 mg/L, with the highest content observed in barley malt + barley beer samples, and the lowest in wheat malt + wheat. The latter case is associated with a low original extract content in the wort sample, as confirmed by other authors [74].

The high content of soluble nitrogen compounds with thiol groups was observed in wheat malt + wheat beer samples (70.2 μM/L), while in other compounds, it varied in the range of 12.3–33.1 μM/L. This agrees with the data of other authors reporting on the structure of protein compounds responsible for foam quality [75,76,77].

When comparing the amounts of iso-α-humulone in different beer samples according to Table 11, it can be observed that wheat malt + wheat beer samples exceed the barley malt sample by 80% according to this indicator, which is due to the difference in the original extract content in the wort. Considering the same hop dosage and different iso-α-humulone levels in beer samples (17.9–25.3 mg/L), we can assume the influence of organic compounds of different adjuncts and fermentation conditions.

The isoxanthohumol content, a hop product prenylflavonoid, also differs in the beer samples. The highest content of this substance refers to wheat malt + wheat beer samples (1.39 mg/L). The level of isoxanthohumol in barley malt, barley malt + barley, barley malt + rice and barley malt + wheat beer samples varies in a narrow range of 0.94–1.04 mg/L, and in the barley malt + corn beer samples, its content is the lowest—0.44 mg/L, which is also the reason for studying the influence of the cereal organic molecules structure on the isoxanthohumol content.

The content of catechins, quercetin, and riboflavin in beer samples varies in the range of 3.0–12.36 mg/L, 0.11–0.66 mg/L, and 0.28–5.26 mg/L, respectively, and is determined by the type of adjuncts with 5% replacement of malt.

Melanoidin content and color in beer samples do not correlate directly with each other, since other organic compounds are also involved in the development of beer color [78].

Based on the data in Table 11, the correlation between iso-α-humulone and organic substances, as well as between isoxanthohumol and beer substances was investigated.

Pairwise correlation coefficients showed that there was a strong relationship between iso-α-humulone and soluble nitrogen compounds (R = −0.85). Figure 3a shows the relationships between organic compounds and iso-α-humulone in beer samples with adjuncts (R = 0.80–0.95).

Mathematical correlation analysis revealed a pairwise correlation between isoxanthohumol and quercetin (R = 0.84), riboflavin (R = 0.84), and melanoidins (0.87). Figure 3b shows the relationships between organic compounds and isoxanthohumol in beer samples with adjuncts (R = 0.81–0.98).

Figure 3 shows the complex interactions of hop substances and the plant matrix of beer, but the relationship with a variety of raw materials, in our opinion, is most responsible for soluble nitrogen compounds, since this indicator refers to the species characteristic of the beer’s proteome.

Figure 4 shows the taste characteristics of beer samples from different adjuncts.

According to Figure 4, the beer samples mostly differed by the descriptor “Smoky tone” and “Floral tone”, which is formed by phenolic compounds, the sources of which are amino acids of raw materials, bioconverted by yeast, and converted into phenolic compounds [79]. The malty tone was the least intense in barley malt + rice beer samples, which is explained by the decrease in amino acid content according to other authors [80].

In Table 11 and Figure 4, the data were subjected to mathematical analysis with respect to the beer flavor descriptors and organic compounds relationship, and the data are shown in Table 12 and Table 13.

In Table 12, the data indicate that a direct strong relationship between the flavor fullness and foam stability descriptors is not found; however, the «smoky» tone correlates with the thiol nitrogen, iso-α-humulone, and quercetin content (R > 0.8). Pairwise correlation coefficients (Table 13) indicate a close relationship between thiol nitrogen compounds, iso-α-humulone, β-glucan, riboflavin, and quercetin on the completeness of taste and stability of beer foam, which is confirmed in part by other authors [75,81,82]. However, no data on the effect of riboflavin and quercetin were found. The only assumption is that since thioredoxins, including thiol groups [83], are responsible for the stability of the foam, respectively, then antioxidants (riboflavin and quercetin) are responsible for the stability of the thiol S-S groups. On the other hand, it was reported that kaempferol, quercetin, and myricetin interacted with nitrogenous compounds via hydroxyl substituents in the B and C rings (positions 3, 3′, 4′, 5′) [84].

With regard to the descriptor “smoke tone”, in addition to the influence of phenolic compounds, the influence of soluble nitrogen and β-glucan was noted, which suggests the importance in the perception of the density of the “body” of beer by the tasters’ receptors, which affects the degree of brightness or perceptibility of the descriptor [82].

## 3. Discussion

By evaluating the contribution of organic compounds and their mutual influence on each other at the kettle hopping stage, it can be observed that there is a relationship between the type of adjuncts introduced to replace part of the malt (5% in a particular case).

Considering the change in bittering and phenolic components, it can be seen that iso-α-humulone correlates with the amount of extractive substances, namely, the extractable and isomerizable hop resin is bound by covalent bonds with organic compounds of different molecular weights that comprise the wort proteome, primarily with LTP and Z peptones [85].

The catechins and quercetin content correlated with the type of adjuncts, i.e., showed the ability of barley malt enzymes to hydrolyze the plant matrix, which differs in free and conjugated phenolic compounds (phenolic acids, residual sugars, etc.) in its structure [86].

A direct effect of catechins on the wort color characteristics has been shown. Meanwhile, melanoidins and riboflavin indirectly influence the wort color index through catechins, which is confirmed in the works of other authors [78].

The decrease in dynamics of the organic compounds content during fermentation was not only due to the biocatalytic processes, but also to the difference in composition of nitrogenous and other compounds. Wheat beer, consisting of wheat malt and 5% wheat, showed the most significant dynamic decrease in β-glucan content. It is known that the β-D-glucan of cereals consist of a linear polymer composed of D-glucopyranosyl units connected by isolated ß-D-(1→3) glycosidic bonds and sets of ß-D-(1→4) glycosidic bonds [67]. Wheat contains less β-glucan in the endosperm wall (up to 25%) compared to barley (up to 80%); however, the structure of wheat has a complex relationship with the arabinoxylan complex, which includes ferulic acid and peptide residues [74]. Malting and mashing conditions promote the extraction of more complex non-starch polysaccharides during the technological process, as confirmed by us and other authors [74].

The usage of barley malt with another adjunct affects the process of clarification of the fermented wort, and the decrease in dynamics follows the chain toward a decrease in the intensity of colloid formation from the cereal type: Wheat→ rice→ barley→ corn, which correlates with the structure and localization of β-glucan [74,87,88].

The dynamics of the change in thiol-containing nitrogen compounds is important in terms of predicting foam stability and the influence of various factors on foam structure. It is interesting to note the grouping of samples according to the rate and nature of the reduction in thiol-containing nitrogen concentration. It was shown that the replacement of 5% barley malt with corn, barley, and rice promotes a strong reduction in thiol-containing nitrogen during the main fermentation (during the first 7 days of fermentation). The difference in the dynamics of thiol nitrogen changes in barley malt beer with adjuncts and wheat beers indicates the special reactivity, i.e., the structure and location of the active sites of amino acid sequences and groups of wheat raw materials (malted and whole wheat cereals) [77]. In terms of the comparative characterization of specific foam structures with thiol-containing amino acids in their composition (LTP1, Z4, Z7-proteins), the corn, barley, and rice proteins are similar in their affinity for binding to organic compounds and differ slightly in their affinity for agglomeration with lipids based on the size of active cavities in the structure, which distinguishes them from the structural and functional properties of LTP1-proteins of wheat [89].

The organic compounds’ influence on colloid formation in accordance with the cereal type helped in clarifying the mathematical analysis. It is shown that in samples with 5% of adjuncts, along with barley and wheat malts, there is a functional correlation between thiol nitrogen and quercetin. In barley malt beer, the correlation coefficient is (0.99), barley malt and barley/rice is (0.99), barley malt and corn is (−0.59), barley malt and wheat is (0.92), and wheat malt and wheat is (0.92). The beer made of 100% malt had correlation coefficients at the level of (R = 0.86–0.99) between nitrogen with thiol groups and the organic compounds in question, while the malt wheat + wheat beer samples had correlation coefficients at the level of (R = 0.92–0.99) and (R = 0.65–0.95), respectively, which suggests the participation of uncounted organic compounds in the connection between quercetin, catechins, iso-α-humulone, and nitrogen compounds with thiol groups.

The study revealed the mutual influence of changes in phenolic compounds on thiol nitrogen in all beer samples, and in particular, the influence of iso-α-humulone and catechin on thiol nitrogen content, which served the purpose of studying the pairwise correlation between these indices in different beer samples.

Iso-α-humolone, an isomerization product of hop α-humol, causes beer bitterness and stabilizes the foam structure [85,90]. The relationship between changes in the content of isomerized resin and catechin can be attributed to a single source of origin (hops) [91]. In addition, catechins are known to inhibit the activity of metalloproteinase enzymes, i.e., they actively bind to the protein molecules [92] present in the structure of beer. This may be due to preferential binding to beer foam proteins at iso-α-humulone binding sites.

The source of riboflavin in beer is malt, which exists in the structure of an aporiboflavin-binding protein and contributes to the removal of riboflavin from beer [93]. Riboflavin’s chemical composition is 7,8-dimethyl-10-ribityl-isoalloxazine, which consists of a flavin-isoalloxazine ring associated with a sugar side chain, ribitol [94]. Riboflavin is resistant to temperature, but not to light. Riboflavin in the form of flavinmonophosphate plays a key role as a cofactor in oxidation and reduction reactions [95] and is active in terms of oxygen absorption. It has been shown that in the presence of quercetin, the antioxidant function of riboflavin increases, i.e., quercetin acts as a synergistic compound [96]. Therefore, to activate riboflavin into flavin monophosphate, it is necessary to influence riboflavin kinase, the content of which is stated by the authors in all cereals [97], and this content decreases due to technological actions (milling). For example, milling wheat flour reduces riboflavin levels by 64%, rice flour by 67%, and corn flour by 56% [98]. It can be assumed that the presented sequence of the decrease in the rate of association of quercetin and riboflavin depends on the concentration of riboflavin or riboflavin kinase in beer from a particular cereal. Moreover, quercetin can bind to protein compounds differently depending on the nature of the protein. Therefore, it was shown that the rice protein matrix had a higher affinity for quercetin compared to the almond protein matrix [99]. It has been established that the reactivity of phenolic compounds with respect to grain proteins correlates with the number of hydroxyl groups and their position in the structure of phenolic compounds, namely, the distribution of the external electronic charge [100].

On the one hand, changes in the bitterness of hop products revealed a lack of correlation with beer organic compounds, and on the other hand, revealed the influence of soluble nitrogen on adjuncts. It is known that the beer’s soluble nitrogen includes the concept of foam peptones (LTP1, Z4,7), peptides and amino acids of raw materials, and sugars formed during fermentation [79]. The connection between the nitrogenous compounds of foam and the bittering resins of hop products has already been discussed [85,90]. The interactions of isomerized resins with amino acids and peptides are confirmed by other authors [101], who describe the occurrence of ionic, ion-dipole, and hydrophobic interactions.

Since phenolic compounds interact with many classes of organic compounds that comprise the beer’s “body”, they consequently affect flavor descriptors. The study confirmed that different phenolic compounds indirectly affect the fullness of flavor, foam quality, and some tones. As already mentioned, the structure of the plant matrix of cereals not only determines the configuration and properties of the nitrogenous compounds, but also the non-starch polysaccharides. Therefore, the degree of influence of thiol nitrogen in combination with iso-α-humolone, β-glucan, riboflavin, and quercetin on flavor descriptors significantly affects tasters’ evaluation of beer quality.

## 4. Materials and Methods

### 4.1. The Research Materials

Barley malt (Russia), wheat malt (Germany), and adjuncts (barley, rice, corn, wheat) were mixed in a ratio of 95:5 by weight and mixed with water in a ratio of 1:4, and then mashed using the infusion method. Hopping was carried out with one hop type variety “Tettnanger” (Germany), and the kettle boiling wort duration was 60 min. Fermentation was carried out with brewer’s yeast *S. cerevisiae*. General stage of fermentation was at a temperature of (9 ± 2) °C for 7 days, and post-fermentation stage was at (3 ± 2) °C for 14 days. Beer production was carried out at the «Easy Drew» pilot brewery (Russia), filtered and stored at temperature (4 ± 2) °C and air humidity W ≤ (75 ± 2)% before the study. The analyzed beers included six samples, in which the characteristics are represented in Table 14.

### 4.2. The Research Methods

#### 4.2.1. Chemicals

All reagents and standards were of analytical grade. Quercetin, isoxanthohumol, riboflavin, and catechin standards were from Sigma-Aldrich (St. Louis, MO, USA) with a purity ≥ of 99%. Potassium dihydrogen phosphate (KH_2_PO_4_), acetonitrile, acetic acid, orthophosphoric acid (H_3_PO_4_), and ammonium dihydro-phosphate (NH_4_H_2_PO_4_) were purchased from Galachem (Moscow, Russia).

Sulfuric acid, boric acid, hydrochloric acid (HCl), ethanol, isooctane, 5,5′-dithiobis [2-nitrobenzoic] acid, and sodium bicarbonate (Na_2_CO_3_) were purchased from Limited liability company “Reatorg” (Moscow, Russia).

Chemicals for determination of β-glucan content were purchased from Megazyme Int. (Lansing, MI, USA).

Bidistilled prepared water was used in the determinations.

#### 4.2.2. Determination of Original Extract and Alcohol Content

To determine the original extract and alcohol content, the 2.13.16.1 standard MEBAK^®^ method was used [102].

#### 4.2.3. Determination of Nitrogen Compounds

To determine the common amount of soluble nitrogen, the Kjeldahl method (EBC Method 4.9.3) was used [103].

#### 4.2.4. Determination of Soluble Nitrogen with Thiol Groups Mass Concentration

We employed Ellman’s method for determining the mass concentration of nitrogen with thiol groups. A total of 3 mL of protein solution from the sample, 2 mL of 0.2 M phosphate buffer (pH 8), and 5 mL of distilled water (sample A) were added to a 20 mL test tube (sample A). Then, 10 mM of Ellman’s reagent was prepared as follows: 37 mg of 5,5′-dithiobis [2-nitrobenzoic] acid was dissolved in 10 mL of 0.1 M potassium phosphate buffer, with pH 7.0, and then stirred. Thereafter, 15 mg of sodium bicarbonate was added to the resulting solution and mixed again. Next, 3 mL of sample A was mixed with 0.02 mL of Ellman’s reagent, with using a micropipette. The sample optical density was measured on a spectrophotometer DR 3900 (HACH-LANGE, GmBH, Berlin, Germany) at a wavelength of 412 nm after 3 min of exposure [104].

#### 4.2.5. Determination of Iso-α-Humulone Mass Concentration

We employed the EBC Method 9.47 for the determination of the mass concentration of iso-α-humulone [105].

#### 4.2.6. Determination of Catechin Mass Concentration

The determination of the catechin mass concentration was carried out using the high-performance liquid chromatography method, with an “Agilent Technologies 1200” LC system (“Agilent Technologies”, Santa Clara, CA, USA) equipped with a diode array detector. HPLC system was equipped with a fitted column Supelco C18 150 × 4.6 mm 5 μm (Thermo, Waltham, MA, USA), with wavelength of 280 nm. The samples and all standard solutions were injected at a volume of 10 μL in a reversed-phase column at 25 °C. HPLC mobile phase was prepared as follows: Solution A: 50 mM of NH_4_H_2_PO_4_ + 1.0 mL of orthophosphoric acid dissolved in 900 mL of HPLC grade water. The volume was comprised of 1000 mL with water and the solution was filtered through 0.45 μm membrane filter and degassed in a sonicator for 3 min. Solution B: Acetonitrile mobile phase was carried out using gradient elution at 1 min, 5% B; at 10 min, 15% B; at 10 to 45 min, 40% B; at 45 to 55 min, 98% B, and at 55 to 60 min, 5% B. The mobile phase flow rate was 1.2 mL/min and the injection volume was 10 μL [106].

#### 4.2.7. Determination of Quercetin Mass Concentration

The determination of the quercetin and rutin mass concentration was carried out using the high-performance liquid chromatography method, with an “Agilent Technologies 1200” LC system (“Agilent Technologies”, Santa Clara, CA, USA) equipped with a diode array detector. HPLC system was equipped with fitted Luna 5 u C18 (2) 250 × 4.6 mm 5 μm (Phenomenex, Torrance, CA, USA) column with wavelength of 290 nm. The samples and all standard solutions at a volume of 20 μL were injected into a reversed-phase column at 25 °C. The mobile phase was 2% acetic acid solution (A) and acetonitrile solution (B) with the ratio (A:B—70:30). The eluent flow rate was 1.5 mL/min [107].

#### 4.2.8. Determination of Isoxanthohumol Mass Concentration

A high-performance liquid chromatography method using “Agilent Technologies 1200” LS system (“Agilent Technologies”, Santa Clara, CA, USA) equipped with a diode array detector was applied to determine the isoxanthohumol mass concentration. HPLC system was equipped with fitted Kromasil C18 150 × 4.6 mm 5 μm (Supelco, Bellefonte, PA, USA) column with wavelength of 290 nm. The samples and all standard solutions at a volume of 10 μL were injected into a reversed-phase column at 25 °C. The mobile phase was acetonitrile solution (A), water (B), and orthophosphoric acid solution (C) with the ratio (A:B:C—40:60:0.1). The eluent flow rate was 1 mL/min [108].

#### 4.2.9. Determination of the Mass Concentration of β-Glucan

To quantify the mass concentration of β-glucan, the standard fermentation method was used (8.13.1) [109].

#### 4.2.10. Determination of the Beer’s Color

To determine the color of beer, the EBC method (EBC Method 9.6) was used [110].

#### 4.2.11. Determination of the Mass Concentration of Riboflavin

The determination of the riboflavin mass concentration was carried out using the high-performance liquid chromatography method, with Agilent Technologies 1200 LC system (Agilent Technologies, Santa Clara, CA, USA) equipped with a diode array detector. HPLC system was equipped with fitted column NanoSpher WSVitamins HILIC 250 × 4.6 mm, 5 μm (Rokland, ON, Canada) with wavelength of 260 nm. The samples and all standard solutions were injected at a volume of 10 μL in a reversed-phase column at 25 °C. The HPLC mobile phase was prepared as follows: Acetonitrile of 50 mM of NH_4_H_2_PO_4_-H_3_PO_4_ was mixed in the ratio of 70:30:0.3. The resulting solution was filtered through a 0.45 μm membrane filter and degassed in an ultrasonic apparatus for 10 min. The mobile phase was carried out using the isocratic elution. The mobile phase flow rate was 1.1 mL/min [93].

#### 4.2.12. Determination of the Mass Concentration of Melanoidins

Wort (beer) melanoidins were extracted using the ethanol dissolution method. A total of 30 mL of wort (beer) was mixed with 100 mL of 10% ethanol (*v*/*v*), left for 20 h (overnight) to extract melanoidins at 4 °C, then the solution was centrifuged at 5000 rpm for 10 min on centrifuge Sigma 2-16KHL (Sigma, Darmstadt, Germany), after which ethanol was added to the supernatant to a 65% (*v*/*v*) final concentration and left for 12 h at 4 °C to dissolve the melanoidins. Next, ethanol was evaporated on a rotary evaporator Advantage G3 ML (Heidolf, Schwabach, Germany) before centrifugation. Then, the supernatant was defatted by mixing with acetone to precipitate melanoidins and the latter was isolated by centrifugation in a centrifuge Sigma 2-16KHL (Sigma, Darmstadt, Germany). Next, melanoidins were resuspended in distilled water (0.6 mg/ml). The optical density of solutions was determined at 420 nm on a spectrophotometer DR 3900 (HACH-LANGE, GmBH, Berlin, Germany) [111].

#### 4.2.13. Organoleptic Evaluation of Beer Samples by Descriptors

The organoleptic analysis was carried out by a professional group of researchers, consisting of 10 people on a 5-point scale according to the characteristic taste descriptors selected. Five points indicate a strong descriptor shade, 4 points indicate a well-developed descriptor shade, 2 points indicate a slightly visible descriptor shade, and 1 point indicates a subtle descriptor shade. The results obtained were summarized and the average score was recorded.

#### 4.2.14. Statistical Analysis

Statistical analysis was performed in five replicates. Descriptive statistics were performed and values are expressed as mean ± standard deviation (SD). In the studies, the Student-Fisher method was used, as a result of which multivariate models of the correlation-regression dependence of the studied parameters were obtained. The reliability limit of the obtained data (*p* ≥ 0.95) was considered to assess various factors affecting the content of polyphenols in all studies. Statistical data were processed by the Statistics program (Microsoft Corporation, Redmond, WA, USA, 2006).

## 5. Conclusions

The article examined the effect of the adjuncts in organic compounds, including phenolic compounds, on the main indicators of hop wort, fermented wort (beer) during fermentation. It was shown that the change in the wort’s phenolic complex correlates with extractive compounds, and during fermentation, it is determined by the adjunct’s proteome. Mathematical analysis showed the effect of quercetin, catechin, and iso-α-humulone on the dynamics and magnitude of the decrease in nitrogen compound content with thiol groups. The effect of quercetin on the dynamics of the reduction in riboflavin levels in beer samples, and the effect of the adjuncts in plant matrix on this process were shown. The reason for the different levels of iso-α-humulone in beer samples with adjuncts has been established, which lies in the specific structure of the nitrogen compound fractions. It was found that the formation of the descriptors “flavor fullness” and “foam stability” through pairwise correlation coefficients are related by thiol nitrogen, iso-α-humulone, β-glucan, riboflavin, and quercetin, while the descriptor “smoky tone” is formed through the relationship between thiol nitrogen compounds, iso-α-humulone, and quercetin. Therefore, the behavior of phenolic compounds in the final stages of beer production is determined by the cereal grain proteome and affects the beer taste and color characteristics, as well as its persistence.

## Figures and Tables

**Figure 1 molecules-28-02295-f001:**
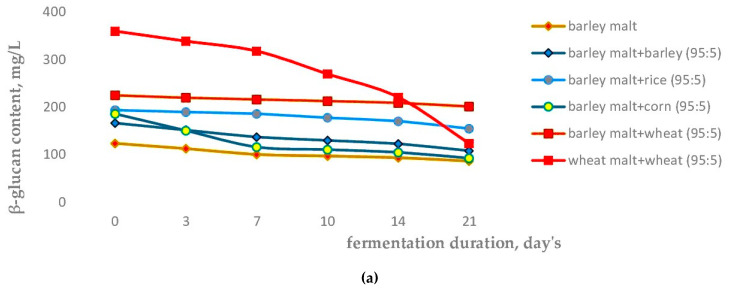
The content of β-glucan (**a**), soluble nitrogen compounds with thiol groups (**b**), catechins (**c**) and iso-α-humulone (**d**), quercetin (**e**), changes in dynamics during wort sample fermentation of different adjuncts.

**Figure 2 molecules-28-02295-f002:**
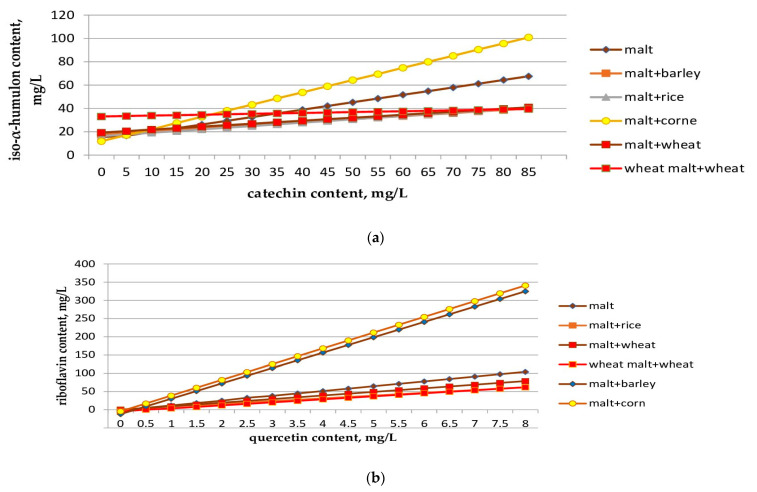
The change in the dynamics of iso-α-humulone from catechin (**a**) and riboflavin from quercetin (**b**) during fermentation.

**Figure 3 molecules-28-02295-f003:**
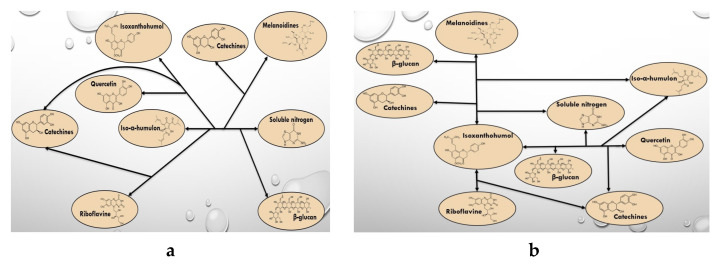
The organic compounds influence on iso-α-humulone (**a**) and isoxanthohumol (**b**) content of different adjuncts of beer samples.

**Figure 4 molecules-28-02295-f004:**
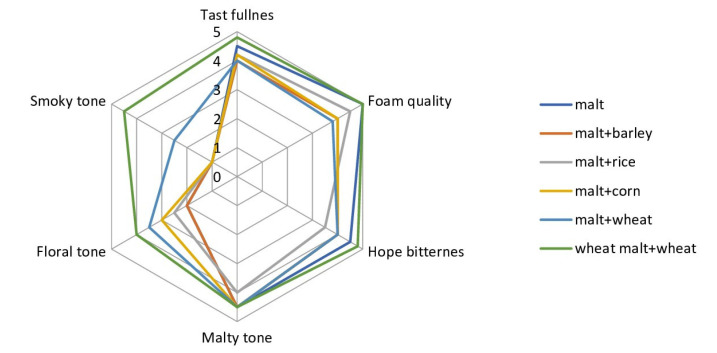
The beer samples taste profiles.

**Table 1 molecules-28-02295-t001:** Phenolic compounds of native cereals.

Phenolic Compounds	Content in Adjuncts (μg/g)			
	Barley [19,20,21]	Rice [22,23,24,25,26,27]	Maize [28,29,30,31,32,33,34,35]	Wheat [36,37,38,39,40]
Phenolic acids				
Gallic	10.10 ± 0.26	4.7 ± 1.6	28.6 ± 2.30	21.0 ± 2.50
Protocatechuic	40.88 ± 5.44	9.9 ± 0.5	41.89 ± 4.55	nf
Chlorogenic acid	50.56 ± 0.84	14.8 ± 0.7	1.56 ± 0.01	27.90 ± 1.0
2,4-Dihydroxybenzoic	4.86 ± 0.92	0.49 ± 0.35	0.24 ± 0.01	39.97 ± 0.41
Vanillic	5.75 ± 1.71	5.1 ± 0.3	1.09 ± 0.01	6.18 ± 0.66
Syringic	4.56 ± 0.35	5.4 ± 0.3	32 ± 3	2.44 ± 0.16
p-coumaric	4.31 ± 0.75	40.0 ± 2.0	6.5 ± 0.4	35.0 ± 3.40
Ferulic	7.22 ± 1.18	34.9 ± 1.7	1748 ± 12	402.10 ± 11.0
Salicylic	7.68 ± 0.33	nd **	nd	nf
benzoic	3.16 ± 1.46	nd	nd	nf
O-coumaric	9.29 ± 0.44	nd	71.0 ± 16	nf
Veratric	5.11 ± 1.36	nd	nd	nd
caffeic	64.4 ± 0.80	6.5 ± 0.3	0.27 ± 0.01	31.83 ± 0.30
sinapic	0.9 ± 0.03	5.9 ± 0.3	132 ± 0.23	3.14 ± 0.14
Flavonoids				
Catechin	46.94 ± 2.72	13.7 ± 5.1	1.68 ± 0.17	0.008 ± 0.0008
Naringin	3.05 ± 0.76	nf	+ ***	nf
Hesperidin	nf *	nf	nf	nf
Myricetin	2.68 ± 1.45	3.50 ± 0.50	nf	nf
Quercetin	12.21 ± 0.24	19.6 ± 0.20	1.58 ± 0.05	24.3 ± 0.64
Naringenin	5.32 ± 0.10	nf	14.8 ± 0.53	7.11 ± 0.45
Kaempferol	9.58 ± 0.46	6.70 ± 0.26	224.0 ± 9.8	6.0 ± 0.32
Rutin	10.32 ± 1.12	13.8 ± 0.81	2.74 ± 0.10	41.7 ± 4.60
apigenin	nf	12.3 ± 1.27	+	59.8 ± 14
tricine	nf	7.3 ± 0.53	nf	nf
luteoline	nf	3.0 ± 0.28	+	2.56 ± 0.12
apimysine	nf	nf	+	nf
Anthocyanins				
Cyanidin	0.66 ± 0.23	nf	nf	nf
Cyanidin-3-glucoside	0.91 ± 0.05	8.47 ± 0.49	409.7 ± 40.2	0.741 ± 0.108
Cyanidin 3-rutinoside	nf	52.0 ± 1.44	nf	0.096 ± 0.003
Delphinidin	1.06 ± 0.40	nf	nf	nf
Delphinidin-3-glucoside	0.56 ± 0.76	0.8 ± 0.05	nf	nf
Malvidin-3-glucoside	0.16 ± 0.18	nf	nf	nf
Pelargonidin-3-glucoside	0.08 ± 0.09	nf	135.9 ± 15.0	nf
Peonidin	0.12 ± 0.11	5.6 ± 0.35	nf	nf
Peonidin-3-glucoside	nf	nf	nf	1.231 ± 0.175
Petunidin	0.40 ± 0.09	nf	nf	nf
Petunidin-3-glucoside	0.32 ± 0.24	nf	152.0 ± 16.0	0.045 ± 0.001

nf *—value not found in literature; ** nd—value not determined by researcher; *** +—compound is qualitatively determined.

**Table 2 molecules-28-02295-t002:** The content of phenolic compounds in beer samples.

Phenolic Compounds	Content in Beer Samples with Adjuncts (mg/L)				
	100% Malt	30% Barley	30% Rice	30% Maize	30% Wheat
2,4-Dihydroxybenzoic	0.59 ± 0.06	0.52 ± 0.01	0.49 ± 0.05	0.45 ± 0.04	0.63 ± 0.06
Vanillic	1.41 ± 0.01	1.56 ± 0.02	0.97 ± 0.01	1.06 ± 0.01	1.48 ± 0.01
Syringic	0.52 ± 0.05	0.32 ± 0.03	0.28 ± 0.03	0.31 ± 0.03	0.40 ± 0.04
p-coumaric	2.79 ± 0.03	1.60 ± 0.02	1.85 ± 0.02	2.50 ± 0.03	1.37 ± 0.01
Ferulic	3.52 ± 0.04	2.55 ± 0.03	2.46 ± 0.02	1.76 ± 0.02	2.73 ± 0.03
Flavonoids					
Catechin	2.6 ± 0.03	2.01 ± 0.02	1.25 ± 0.01	0.87 ± 0.09	0.97 ± 0.03
Quercetin	1.5 ± 0.1	nd	nd	nd	4.93 ± 0.06
Rutin	5.8 ± 0.2	nd	nd	nd	0.34 ± 0.02
Proanthocyanidins					
monomers (catechin + epicatechin)	4.0				
dimers (procianines + prodelfinidins)	11.0				
trimers	3.0				
4–6 trimers	4.0				

nd—value not determined by researcher.

**Table 3 molecules-28-02295-t003:** The influence of adjuncts on beer attributes.

Non Malted Adjunct‘s Type (Less 20%)	Effect on Sensory/ Nutritional Properties	References
Wheat	More clarity beer, higher alcohol content and less foam retention, organoleptic parameters have been changed: a more pronounced grain smell, a denser body, less astringency and bitterness, including due to a decrease in the phenolic compound’s content and a change in the nitrogenous compound’s composition	[51,54]
Barley	Lighter color, more noticeable bitter taste, increased astringency, which is associated, among other things, with a decrease in the phenolic compound’s number and changes caused by amino acids set that turn into alcohols during fermentation	[51,55,56,57]
Maize	Gives a specific aroma to beer like corn due to the presence of 6-acetyltetrahydropyridine, 2-acetyl-1-pyrroline and its analogue 2-propionyl-1-pyrroline, a decrease in color intensity due to a decrease in the phenolic compound’s content, antioxidant activity, an increase in the peptides and free amino nitrogen level	[58,59,60,61,62]
Rice	Not abundant foam, flat taste, few free amino acids, difficult to hydrolyze protein, which affects the volatile compound’s profile and the foam quality	[63,64,65]

**Table 4 molecules-28-02295-t004:** The characteristics of wort samples.

Sample Number	The Content in Samples, mg/L, Reliability Limit *p* < 0.05				
	Original Extract, °P	β-Glucan (Gl)	Soluble Nitrogen (SN)	Soluble Nitrogen with Thiol Groups (SNTG), μM/L	Iso-α-Humulone (IBU) (IH)
1BMW	(12.6 ± 1.0) *	124.1 ± 8.7	817.2 ± 33	396.0 ± 20	26.1 ± 0.11
2BMBW	11.0 ± 1.0	166.8 ± 11.7	927.3 ± 37	812.3 ± 40	29.6 ± 0.12
3BMRW	13.0 ± 1.0	194.0 ± 13.6	1095.3 ± 44	236.1 ± 12	25.9 ± 0.11
4BMCW	12.7 ± 1.0	186.2 ± 13.0	1020.5 ± 40	989.5 ± 50	33.9 ± 0.15
5BMWW	12.4 ± 1.0	225.0 ± 15.7	1025.3 ± 40	216.2 ± 11	27.9 ± 0.12
6WMWW	10.2 ± 1.0	360.0 ± 25.2	654.1 ± 26	288.4 ± 14	28.8 ± 0.12

*—each value represents the mean of five independent experiments (±SD).

**Table 5 molecules-28-02295-t005:** The characteristics of wort samples, which form color.

Sample Number	The Content in Samples, mg/L, Reliability Limit *p* < 0.05				
	Catechin (Ct)	Quercetin (Qv)	Melanoidin (Mel)	Riboflavin (Rf)	Color, °EBC
1BMW	(18.6 ± 1.9) *	0.28 ± 0.03	126.0 ± 12.0	0.46 ± 0.005	12.0 ± 0.36
2BMBW	38.4 ± 3.8	0.35 ± 0.03	125.0 ± 12.0	2,86 ± 0.028	11.0 ± 0.33
3BMRW	24.75 ± 2.5	0.43 ± 0.04	185.0 ± 18.0	4.33 ± 0.043	14.0 ± 0.42
4BMCW	21.1 ± 2.0	0.14 ± 0.01	136.0 ± 13.6	1.55 ± 0.015	25.0 ± 0.75
5BMWW	24.75 ± 2.5	0.39 ± 0.04	170.0 ± 17.0	3.87 ± 0.039	13.25 ± 0.40
6WMWW	81.7 ± 8.2	0.98 ± 0.10	225.0 ± 22.0	3.72 ± 0.037	65.0 ± 1.95

*—each value represents the mean of five independent experiments (±SD).

**Table 6 molecules-28-02295-t006:** Pairwise correlation coefficients between organic components.

	Pairwise Correlation Coefficient, Reliability Limit *p* < 0.05			
	Color (y)	Ct (x_1_)	Mel (x_2_)	Rf (x_3_)
Color (y)	1	0.90	0.75	0.23
Ct (x_1_)	0.90	1	0.72	0.40
Mel (x_2_)	0.75	0.72	1	0.72
Rf (x_3_)	0.23	0.40	0.72	1

**Table 7 molecules-28-02295-t007:** Samples of pairwise correlation coefficients (reliability limit *p* < 0.05).

-	NTG (Y)	Ct (x1)	Qv (x2)	IG (x3)
	barley malt beer Y = −976.9757 − 6.163X_1_ + 56.7859X_2_ + 14.4257X_3_			
NTG (Y)	1	0.86	0.99	0.91
Ct (x_1_)	0.86	1	0.90	0.99
Qv (x_2_)	0.99	0.90	1	0.94
IG (x_3_)	0.91	0.99	0.94	1
	barley malt and barley beer Y= −2211.4679 − 11.9219X_1_ + 125.3206X_2_ − 650.3287X_3_			
NTG (Y)	1	0.79	0.98	0.84
Ct (x_1_)	0.79	1	0.90	0.98
Qv (x_2_)	0.98	0.90	1	0.94
IG (x_3_)	0.84	0.98	0.94	1
	barley malt and rice beer Y = −569.4632 − 0.3169X_1_ + 31.4308X_2_ − 1.5104X_3_			
NTG (Y)	1	0.78	0.99	0.96
Ct (x_1_)	0.78	1	0.79	0.91
Qv (x_2_)	0.99	0.79	1	0.96
IG (x_3_)	0.96	0.91	0.96	1
	barley malt and corn beer Y = 171.3174 − 7.8932X_1_ + 23.24X_2_ − 5813.8946X_3_			
NTG (Y)	1	−0.63	−0.59	−0.75
Ct (x_1_)	−0.63	1	0.99	0.97
Qv (x_2_)	−0.60	0.99	1	0.98
IG (x_3_)	−0.75	0.97	0.98	1
	barley malt and wheat beer Y = −192.7518 + 6.1741X_1_ + 9.0969X_2_ + 4.9172X_3_			
NTG (Y)	1	0.98	0.92	0.99
Ct (x_1_)	0.98	1	0.84	0.97
Qv (x_2_)	0.92	0.84	1	0.92
IG (x_3_)	0.99	0.97	0.92	1
	wheat malt and wheat beer Y = −1838.8135−2.5459X_1_ + 60.6517X_2_ − 20.2411X_3_			
NTG (Y)	1	0.65	0.92	0.95
Ct (x_1_)	0.65	1	0.90	0.85
Qv (x_2_)	0.92	0.90	1	0.99
IG (x_3_)	0.95	0.85	0.99	1

**Table 8 molecules-28-02295-t008:** Samples of pairwise correlation characteristics (reliability limit *p* < 0.05).

Beer’s Raw Material Type	Pairwise Correlation Coefficient	Elasticity (E), Correlation (R), and Determination (R^2^) Coefficients
barley malt	Ct-IGM/NTG (−1); Ct-Qv/NTG (−0.99); IGM-Qv/NTG (1)	E_ct_ = −0.556; E_igm_ = 8.865; E_qv_ = 0.0263; R = 1; R^2^ = 1
barley malt + barley	Ct-IGM/NTG (1); Ct-Qv/NTG (0.99); IGM-Qv/NTG (−0.62)	E_ct_ = −1.039; E_igm_ = 12.243; E_qv_ = −0.905; R = 1; R^2^ = 1
barley malt + rice	Ct-IGM/NTG (1); Ct-Qv/NTG (−0.79); IGM-Qv/NTG (0.92)	E_ct_ = −0.0577; E_igm_ = 7.288; E_qv_ = −0.00548; R = 1; R^2^ = 1
barley malt + corn	Ct-IGM/NTG (0.98); Ct-Qv/NTG (0.97); IGM-Qv/NTG (0.99)	E_ct_ = −4.229; E_igm_ = 22.162; E_qv_ = −2.523; R = 1; R^2^ = 1
barley malt + wheat	Ct-IGM/NTG (−1); Ct-Qv/NTG (−0.64); IGM-Qv/NTG (0.28)	E_ct_ = 0.799; E_igm_ = 1.687; E_qv_ = 0.012; R = 1; R^2^ = 1
wheat malt + wheat	Ct-IGM/NTG (1); Ct-Qv/NTG (1); IGM-Qv/NTG (0.99)	E_ct_ = −0.369; E_igm_ = 10.974; E_qv_ = −0.085; R = 1; R^2^ = 1

**Table 9 molecules-28-02295-t009:** The content of iso-α-humulone (Y) on catechins (X) characteristics dependence on the beer’s raw material type.

Beer’s Raw Material Type	Correlation Characteristics, Reliability Limit *p* < 0.05	
	Dependency Equation	Correlation (R) and Determination (R^2^) Coefficients
barley malt	Y = 13.5162 + 0.6358X	R = 0.98; R^2^ = 0.96
barley malt + barley	Y = 17.6601 + 0.2628X	R = 0.95; R^2^ = 0.90
barley malt + rice	Y = 16.2796 + 0.2882X	R = 0.83; R^2^ = 0.69
barley malt + corn	Y = 11.8962 + 1.0479X	R = 0.98; R^2^ = 0.97
barley malt + wheat	Y = 19.3289 + 0.2542X	R = 0.83; R^2^ = 0.69
wheat malt + wheat	Y = 33.1081 + 0.0757X	R = 0.90; R^2^ = 0.81

**Table 10 molecules-28-02295-t010:** The content of riboflavin (Y) on quercetin (X) characteristics dependence on the beer’s raw material type.

Beer’s Raw Material Type	Correlation Characteristics, Reliability Limit *p* < 0.05	
	Dependency Equation	Correlation (R) and Determination (R^2^) Coefficients
barley malt	Y = 13.1969X−1.4249	R = 0.95; R^2^ = 0.90
barley malt + barley	Y = 42.1003X−11.9372	R = 0.94; R^2^ = 0.88
barley malt + rice	Y = 7.6103X + 0.9626	R = 0.91; R^2^ = 0.83
barley malt + corn	Y = 43.1731X−4.4028	R = 0.99; R^2^ = 0.99
barley malt + wheat	Y = 9.8901X−0.4176	R = 0.93; R^2^ = 0.86
wheat malt + wheat	Y = 8.3123X−4.7546	R = 0.91; R^2^ = 0.83

**Table 11 molecules-28-02295-t011:** The characteristics of beer samples with different adjuncts.

Measuring Parameters List	The Content in Samples, mg/L, Reliability Limit *p* < 0.05					
	1BMW	2BMBW	3BMRW	4BMCW	5BMWW	6WMWW
Brix, °P	(5.4 ± 0.2) *	4.2 ± 0.1	6.4 ± 0.2	4.3 ± 0.1	7.6 ± 0.2	6.1 ± 0.2
Alcohol content, %vol	5.7 ± 0.1	4.2 ± 0.1	5.5 ± 0.1	4.4 ± 0.3	5.7 ± 0.2	3.7 ± 0.1
β-Glucan content (Gl)	86.9 ± 6.1	108.6 ± 7.6	155.2 ± 10	93.1 ± 6.5	201.7 ± 14	124.1 ± 9
Soluble Nitrogen content (SN)	628.9 ± 25	695.6 ± 97	649.7 ± 59	610.3 ± 85	672.6 ± 94	495.2 ± 69
Soluble Nitrogen with Thiol Groups content (SNTG), μM/L	12.3 ± 0.6	33.1 ± 1.6	12.3 ± 0.6	32.9 ± 1.6	29.8 ± 1.5	70.2 ± 3.5
Iso-α-humulone content (IBU) (IH)	17.9 ± 0.07	19.8 ± 0.08	18.6 ± 0.07	25.3 ± 0.1	21.1 ± 0.08	32.1 ± 0.1
Isoxanthohumol content (IXG)	1.04 ± 0.05	0.94 ± 0.05	0.97 ± 0.05	0.44 ± 0.02	0.99 ± 0.05	1.39 ± 0.07
Catechin content (Ct)	5.45 ± 0.3	3.00 ± 0.2	4.95 ± 0.2	12.36 ± 0.6	4.70 ± 0.2	9.86 ± 0.5
Quercetin content (Qv)	0.21 ± 0.01	0.31 ± 0.02	0.26 ± 0.01	0.11 ± 0.01	0.25 ± 0.01	0.66 ± 0.03
Riboflavin content (Rf)	1.39 ± 0.15	1.24 ± 0.10	3.02 ± 0.3	0.28 ± 0.03	2.21 ± 0.2	5.26 ± 0.5
Melanoidin content (Mel)	106.4 ± 15	136.0 ± 19	134.0 ± 19	35.0 ± 5	150.0 ± 21	166.0 ± 23
Color, °EBC	10 ± 0.3	9.5 ± 0.3	10.5 ± 0.3	13.7 ± 0.4	9.6 ± 0.3	28.0 ± 0.8

*—each value represents the mean of five independent experiments (±SD).

**Table 12 molecules-28-02295-t012:** Pairwise correlation coefficients of different organic compounds of beer samples (reliability limit *p* < 0.05).

Descriptor	Pairwise Correlation Coefficient of Different Organic Compounds and Beer Descriptors							
	Gl	SN	NTG	Ct	IG	IKG	Qv	Rf
Taste fullness	−0.37	−0.58	0.32	0.49	0.49	0.56	0.50	0.54
Foam quality	−0.39	−0.27	−0.05	0.17	0.09	0.59	0.32	0.44
Smoky tone	0.30	−0.60	0.83	0.39	0.84	0.64	0.84	0.78

**Table 13 molecules-28-02295-t013:** Correlation characteristics of different organic compounds of beer samples (reliability limit *p* < 0.05).

Beer’s Descriptor Type	Pairwise Correlation Coefficient	Correlation (R) and Determination (R^2^) Coefficients
Taste fullness	NTG-IG/TF (0.93); Gl-Rf/TF (0.84); Ct-Qv/TF (0.84); NTG-Qv/TF (0.78); Qv-Rf/TF (0.77); IXG-Qv/TF (0.72); Gl-IXG/TF (0.69); SN-Rf/TF (−0.65); SN-NTG/TF (−0.65); Gl-Qv/TF (0.60); SN-IXG/TF (−0.57)	R = 0.86; R^2^ = 0.75
Foam stability	NTG-IG/FS (0.93); NTG-Qv/FS (0.86); Qv-Rf/FS (0.81); IXG-Qv/FS (0.80); Gl-Rf/FS (−0.74); Ct-IG/FS (0.73); SN-NTG/FS (−0.73); IG-Qv/FS (0.70); SN-IG/FS (−0.69)	R = 0.93; R^2^ = 0.86
Smoky tone	IG-IXG/ST (0.926); Ct-IG/ST (0.882); Gl-IG/ST (0.881); Ct-Qv/ST (0.881); SN-Rf/ST (−0.871); Gl-SN/ST (0.86); Ct-IXG/ST (0.847); SN-Qv/ST (0.82)	R = 0.93; R^2^ = 0.86

**Table 14 molecules-28-02295-t014:** The beer samples characteristics.

Sample Code	Raw Material List	Yeast Type	Color	Hopping Technology
1BMW	light barley malt, hop pellets	lager	light	kettle hopping
2BMBW	light barley malt, barley, hop pellets	lager	light	
3BMRW	light barley malt, rice, hop pellets	lager	light	
4BMCW	light barley malt, corn, hop pellets	lager	light	
5BMWW	light barley malt, wheat, hop pellets	lager	light	
6WMWW	light wheat malt, wheat, hop pellets	lager	light	

## Data Availability

Data confirming the published results are presented in this article.

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
