# Peer review of "The Phenolic Compounds’ Role in Beer from Various Adjuncts"

_molecules, 2023, doi:10.3390/molecules28052295_

Round 1

Reviewer 1 Report

In this manuscript, the authors describe the effect the various adjuncts on beer malt, which may influence the constituent organic compounds and taste profile composition of the beer

The authors attempted to study the interactions of the beer phenolic compounds with other biomolecules, and to expand their knowledge on the contribution of the adjunct organic compounds on the beer quality.

The beers sampled originated from a pilot brewery using barley and wheat malts, barley, rice, corn and wheat. The beer samples were assessed with industry-accepted methods (HPLC) and the data were processed by a valid the statistical program.

The study disclosed that at the stage of hoped wort organic compounds structure formation, there was a clear correlation between the content of organic compounds (including the phenolic compounds quercetin, catechins), dry substances and isomerized hop’s bitter resins. It is shown that the riboflavin content increases in all adjunct’s wort samples.

They also established that the changes in ß-glucan and nitrogen with thiol groups during fermentation occurred with different dynamics and depending on the adjunct’s proteome. In addition, they noticed that the greatest decrease in non-starch polysaccharide was observed in wheat beer. They indicated that the  change in isohumulone in all samples was observed at the beginning of fermentation. And that the behavior of catechins, quercetin and isohumulone were to be correlated with nitrogen with thiol group compounds.

They also established that various phenolic compounds were involved in the formation of taste, structure, antioxidant properties of beer in accordance with the structure of various grains, depending on the structure of its proteome.

The authors concluded that obtained experimental and mathematical results allow then to under-stand the intermolecular interactions of the various beer organic using adjuncts.

This manuscript needs a strong revision. The English lexicon needs to be modified. Unfortunately. the content of this manuscript can only be understood by experts in  research in fermentation, with a strong focus on brewing. This is why the authors have to make an effort to explain the successive steps and methodology of brewing, defining the adjuncts, explaining all the unusual terms used throughout………And most of all, not to assume and believe that all the readers of this journal are experts in brewing

For these reasons, this manuscript needs a very strong revision. The following are some  hints of improvements that should be considered by the authors.

1.In your Abstract you wrote the following: The greatest decrease in non-starch polysaccharide was observed in wheat beer, and nitrogen with thiol groups, on the contrary, in all other samples, except for wheat beer.

Query: This is completely incomprehensible!

2.In page 4 of your pdf, line 109,, you have mentioned the term “ non malted Adjunct”

Query: Define what the term “ non malted Adjunct” really means.

What is a beer Adjunct?

You must  add in your text a part describing “Adjunct” and what are the benefits of adding “Adjunct” in your Table 3,

which is entitled as follows: Table 3 shows the flavor attributes according to the adjuncts used.

3.In your Table 3, put a decent space between your non-malted adjunct used.

And when you start a new paragraph use a Capital letter!

4.In page $, line 121, please correct different adjuncts.

5. In page In page 5, line 134.

Query: What is the difference between wheat  malt and wheat wort?

You most probably know, but don’t assume that any reader of your manuscript will know exactly what you mean!

6. In page In Table 5, Table 5, please define what EBC stands for!

7. In  page 7, Table 6, please add a  foot note! under the table explaining what Ct (x1, Mel (x2) and Rf (x3) mean!

8. In page 7, lines 173-174, in your Fig. 1. The content of ß-glucan (a), soluble nitrogen with thiol groups (b), catechines (c) and iso-a-humulon (d), quercetin (e.) changes dynamics during different adjuncts worts samples fermentation.

Query: Please add  method references used to determine this series of analytes.

9.In page 7, lines 177-180, you have written the following:

The change in the content of ß-glucan (Fig. 1a) occurs most significantly in fermented wort from wheat malt and wheat – by 3 times, since the formation of colloidal particles consisting of nitrogenous, phenolic and non-starch carbohydrate compounds occurs, which are sufficiently weighty by molecular weight to remain in equilibrium in the dispersed structure [66].

Query: Can you explain the relationship between the content of ß-glucan and the formation of colloidal particles consisting of nitrogenous, phenolic and non-starch carbohydrate compound. Frankly, this referee does not understand that!

10. In page 8, lines 185-188, you have written the following:

In other words, a lower ß-glucan concentration indicates a more dissolved structure of the cereal components and a lower molecular weight of the non-starch polysaccharide inherent in the fermenting wort, on the one hand, and the type of structure, on the other hand [67,68].

Query as #8, can you explain the relationship between the lower ß-glucan concentration and the dissolved structure?

Dissolved structure of what?

11. Same page 8, line 180. You have written the following: “Changes in soluble nitrogen with thiol groups (Fig. 1b)”

Query: From where these soluble in nitrogen with thiol groups originated? Obviously proteins, but you have to explain it.

12. In page 8, lines  193-194, can you care to explain why there is a  high intensity of nitrogen reduction in the first 7 days of fermentation?

13. In page 11, Table 2, your first row on the left-hand side  says the following:

:” malt; Y=13.1969X-1.4249; R=0.95; R2=0.90, It is ranked the second highest coefficient

And according to your text instead of “malt” it should be  “barley malt-barley beer”

As a general comment when you discuss “malt” you should specify from which grain it originated from.

14. In page 15, line 389, you mention in your discussion;” kettle hopping stage,”, Frankly, this referee is not an expert in beer making, but perhaps a little explanation is necessary here!

15.In page 15, line 404, please correct as shown: “It is known that the ß-D-glucan of cereals consist of a linear polymer composed of D-glucopyranosyl units connected by isolated ß-D- (1→3) glycosidic bonds and sets of ß-D-(1→4) glycosidic bonds [87]. Please don’t mangle the Carbohydrate Nomenclature.

Author Response

The authors are grateful for the work done by the reviewer in proofreading the article.  

Reviewer 2 Report

The article is not suitable for publication, since it outside the scope of the journal and it seems to be have no sound.

Author Response

The authors are grateful to the referee for the time devoted to the materials of the article. 

Reviewer 3 Report

The topic under consideration is relevant, since it studies the interactions of phenolic compounds with other biomolecules, and expands the understanding of the adjunct’s organic compounds contribution and their joint effect on the beer’s quality.

The beers were sampled at a pilot brewery using barley and wheat malts, barley, rice, corn, and wheat, and fermented; beer’s samples were assessed by industry-accepted methods and using instrumental analysis methods (high-performance liquid chromatography methods—HPLC). The obtained statistical data were processed by the Statistics program (Microsoft Corporation, Redmond, WA, USA, 2006). The study showed that at the stage of hoped wort organic compounds structure formation, there is a clear correlation between the content of organic compounds and dry substances, including phenolic com-pounds (quercetin, catechins), as well as isomerized hop’s bitter resines. It is shown that the riboflavin content increases in all adjunct’s wort samples, and most of all with the use of rice - up to 4.33 mg/L, which is 9.4 times higher than the vitamin levels in malt wort. The melanoidin content in the samples was in the range of 125-225 mg/L and it’s levels in the wort with additives exceeded that of the malt wort. Changes in β-glucan and nitrogen with thiol groups during fermentation occurred with different dynamics and depending on the adjunct’s proteome. The greatest decrease in non-starch polysaccharide was observed in wheat beer, and ni-trogen with thiol groups, on the contrary, in all other samples, except for wheat beer. The change in isohumulone in all samples at the beginning of fermentation correlated with a decrease in original extract, and in the finished beer there was no correlation. The behavior of catechins, quercetin and isohumulone has been shown to correlate with nitrogen with thiol groups during fermentation. A strong correlation was shown between the change in isohumulone and catechins, as well as riboflavin and quercetin. It was established that various phenolic compounds were involved in the formation of taste, structure, antioxidant properties of beer in ac-cordance with the structure of various grains, depending on the structure of its proteome. Conclusions: The obtained experimental and mathematical dependences make it possible to expand the understanding of in-termolecular interactions of beer’s organic compounds and take a step towards predicting the quality of beer at the stage of using adjuncts.

It is interesting topic and the manuscript is well organized. However, there still have some issues need to check. The reference should be updated.

1.      The introduction should be compressed and rearranged.

2.      The relationship of phenolic compounds and antioxidant properties should be analyzed (The positive correlation of antioxidant activity and prebiotic effect about oat phenolic compounds. Food Chemistry, 402(2023): 134231.).

3.      Maillard reaction should be considered during fermentation(Dietary polyphenols: regulate the advanced glycation end products (AGEs)-RAGE axis and the microbiota-gut-brain axis to prevent neurodegenerative diseases. Critical Reviews in Food Science and Nutrition. Doi: 10.1080/10408398.2022.2076064.).

4.      The significance analysis needs to be supplemented in the table.

5.      The analyzation of beer quality and β-glucan content should be analyzed(Recent advances of cereal beta-glucan on immunity with gut microbiota regulation functions and its intelligent gelling application. Critical Reviews in Food Science and Nutrition. doi: 10.1080/10408398.2021.1995842.).

6.      The clarity of the picture needs to be improved.

Author Response

The authors express their gratitude to the referee for the comments that arose during the work on the materials of the article. 

Round 2

Reviewer 2 Report

The authors did the corrections wanted and the manuscript could be accepted for publication.

Reviewer 3 Report

accept